# A Compartmental Mathematical Model to Assess the Impact of Vaccination, Isolation, and Key Epidemiological Parameters on Mpox Control

**DOI:** 10.3390/medsci13040226

**Published:** 2025-10-10

**Authors:** Pedro Pesantes-Grados, Nahía Escalante-Ccoyllo, Olegario Marín-Machuca, Abel Walter Zambrano-Cabanillas, Homero Ango-Aguilar, Obert Marín-Sánchez, Ruy D. Chacón

**Affiliations:** 1Unidad de Posgrado, Facultad de Ciencias Matemáticas, Universidad Nacional Mayor de San Marcos, Lima 15081, Peru; pedro.pesantesgrados.1@gmail.com; 2Grupo de Investigación Ciencias Matemáticas para las Ciencias de la Vida, Lima 15081, Peru; 3Departamento de Computer Science, Facultad de Computación, Universidad de Ingeniería y Tecnología—UTEC, Barranco, Lima 15063, Peru; nahia.escalante@utec.edu.pe; 4Departamento Académico de Ciencias Alimentarias, Facultad de Oceanografía, Pesquería, Ciencias Alimentarias y Acuicultura, Universidad Nacional Federico Villarreal, Miraflores, Lima 15074, Peru; omarin@unfv.edu.pe; 5Departamento de Acuicultura, Facultad de Oceanografía, Pesquería, Ciencias Alimentarias y Acuicultura, Universidad Nacional Federico Villarreal, Miraflores, Lima 15074, Peru; azambrano@unfv.edu.pe; 6Programa Académico de Microbiología. Facultad de Ciencias Biológicas, Universidad Nacional de San Cristóbal de Huamanga, Portal Independencia N° 57—Huamanga, Ayacucho 05001, Peru; homero.ango@unsch.edu.pe; 7Departamento Académico de Microbiología Médica, Facultad de Medicina, Universidad Nacional Mayor de San Marcos, Av. Carlos Germán Amezaga 375, Lima 15081, Peru; omarins@unmsm.edu.pe; 8Pathogen Genetics Research Group (PATHO-GEN), OMICS, Lima 15001, Peru; 9Department of Pathology, School of Veterinary Medicine, University of São Paulo, São Paulo 05508-270, SP, Brazil

**Keywords:** mathematical model, monkeypox disease, basic reproduction number, sensitivity analysis, transmission, vaccination

## Abstract

Background: Monkeypox (Mpox) is a re-emerging zoonotic disease caused by the monkeypox virus (MPXV). Transmission occurs primarily through direct contact with lesions or contaminated materials, with sexual transmission playing a significant role in recent outbreaks. In 2022, Mpox triggered a major global outbreak and was declared a Public Health Emergency of International Concern (PHEIC) by the World Health Organization (WHO), prompting renewed interest in effective control strategies. Methods: This study developed a compartmental SEIR-based model to assess the epidemiological impact of key interventions, including vaccination and isolation, while incorporating critical epidemiological parameters. Sensitivity analyses were conducted to examine (1) disease dynamics in relation to the basic reproduction number, and (2) how different parameters influence the curve of symptomatic infections. Real-world continental-scale data were used to validate the model and identify the parameters that most significantly affect epidemic progression and potential control of Mpox. Results: Results showed that the basic reproduction number was most influenced by the recovery rate, vaccination rate, vaccine effectiveness, and transmission rates of symptomatic and asymptomatic individuals. In contrast, the progression of symptomatic cases was highly sensitive to the case fatality rate and incubation rate. Conclusions: These findings highlight the importance of integrated public health strategies combining vaccination, isolation, and early transmission control to mitigate future Mpox outbreaks.

## 1. Introduction

Monkeypox (Mpox) is an emerging zoonotic disease that affects humans and other mammals. It is typically characterized by rash and skin lesions, although it may also present symptoms such as fever, chills, and swollen lymph nodes. The incubation period ranges from 7 to 17 days, with clinical manifestations usually appearing around day 21 [1]. A significant multinational outbreak occurred in May 2022 and was declared a public health emergency of international concern (PHEIC) by the World Health Organization (WHO) in July 2022 [2]. Transmission of Mpox is primarily human-to-human, occurring through direct contact with skin or mucosal lesions, contaminated body fluids, and fomites. Sexual transmission, particularly among men who have sex with men (MSM), has also been identified as a major route [3]. By the end of 2022, approximately 85,000 confirmed cases and 138 deaths had been reported, primarily in Europe, North America, and South America [4].

The causative agent of Mpox is the monkeypox virus (MPXV), a double-stranded DNA virus belonging to the family *Poxviridae* and the genus *Orthopoxvirus*. MPXV is phylogenetically related to the smallpox virus (Variola virus) but exhibits lower lethality and transmissibility [5]. There are two main genetic clades of the virus: Clade I, which originated in Central Africa and is associated with higher virulence and mortality, and Clade II, which originated in West Africa and was responsible for the global outbreak of 2022–2023 [6]. Recent comparative genomic studies have identified mutations associated with the virus’s adaptation to sustained human-to-human transmission, particularly in coding regions related to immunomodulatory proteins and virulence factors [7,8]. This enhanced ability to evade innate and adaptive immune responses, combined with prolonged viral shedding from skin and mucous membranes, contributes to transmission even in the absence of severe symptoms [9]. In August 2024, the WHO declared Mpox a PHEIC for the second time. This outbreak primarily affected the Democratic Republic of the Congo (DRC) and was caused by Clade I, which is associated with different case fatality rates (CFR) [10].

Despite the use of vaccines such as JYNNEOS (MVA-BN) and ACAM2000, both based on vaccinia viruses, several studies have shown that vaccination alone is not sufficient to contain outbreaks [11,12]. Similarly, data from the 2022 epidemic indicate that changes in social behavior and nonpharmaceutical interventions played a key role in reducing case numbers, even before significant vaccination coverage was achieved in high-risk populations [13]. The WHO has also emphasized that containment strategies must include complementary public health measures such as early case detection, isolation of infected individuals, contact tracing, and reducing high-risk behaviors [2]. Moreover, studies have shown that reducing sexual contacts among high-risk populations may be even more effective than mass vaccination, particularly when vaccine coverage is low or when implementation is delayed [14]. Although the 2022 outbreak predominantly affected MSM populations, subsequent reports documented secondary community transmission and the potential for zoonotic reintroductions [15,16]. For this reason, using an extended mathematical model is beneficial, as it can capture broader transmission scenarios, incorporate waning immunity and asymptomatic carriers, and thus provide a generalizable framework to evaluate both current and future epidemic dynamics. Given that Mpox is a zoonotic infection, some modeling approaches have incorporated both human and animal subsystems to capture its endemic dynamics [17,18,19,20]. While such frameworks are particularly relevant for Central and West Africa, where animal reservoirs contribute to sustained transmission, our study focuses on the 2022–2023 outbreaks in non-endemic regions. These were predominantly human-to-human transmission events, for which the animal subsystem could reasonably be neglected. Accordingly, and to ensure parsimony, we restricted our analysis to the human system while validating the model with continental-scale epidemiological data.

Mathematical modeling plays a crucial role in understanding and managing the spread of infectious diseases including Mpox, offering a framework to estimate key epidemiological parameters such as the basic reproduction number (R_0_), the effective reproduction number (R_e_) and the coefficient of determination (R^2^) when compared to real data [13,21,22], and to assess the potential impact of interventions. Compartmental models are particularly useful in infectious disease research due to their simplicity, flexibility, and ability to represent transitions between different stages of infection (susceptible, exposed, infected, recovered). These models can be extended to incorporate latent infection, waning immunity, and reinfection dynamics, making them especially relevant for emerging zoonoses like Mpox. More advanced approaches, including fractional-order models, have captured complex features such as delayed immune responses and “memory effects” from prior exposures [17]. These tools not only support real-time public health decision-making but also allow simulations of multiple scenarios, including vaccination rollouts, behavioral changes, and social interventions.

In response to these challenges, this study develops a compartmental mathematical SEIR-based model formulated as a system of differential equations. The model includes multiple clinical and epidemiological compartments (susceptible, exposed, symptomatic, asymptomatic, isolated, recovered, and vaccinated) and incorporates key parameters such as vaccine efficacy, waning immunity, and the relative infectiousness of asymptomatic carriers. Real-world case data and sensitivity analyses are used to simulate a range of outbreak scenarios and to identify the parameters most critical for epidemic control. This integrative approach aims to provide a more flexible and robust tool for predicting outbreak dynamics and supporting public health decision-making.

## 2. Materials and Methods

### 2.1. Model Building

A compartmental model, formulated as a system of differential equations, was developed to describe Mpox transmission dynamics within a population (N). The model divides the population into seven epidemiological states: susceptible (S), exposed (E), symptomatic infected (I), asymptomatic infected (A), symptomatic infected isolated (Q), recovered (R), and vaccinated (V). The dynamics of disease spread are posed as follows: A susceptible individual (S) may come into contact with a symptomatic infected or asymptomatic individual (I, A, which are the only transmitters considered) and contract the disease. Once contracted, the disease enters an incubation period (E) during which it cannot spread. Subsequently, the individual may develop symptoms (I) or not (A) but is already considered infectious. A portion of symptomatic individuals is placed in isolation (Q), and therefore their ability to transmit the disease is not considered. Finally, all infected individuals may recover (R) or die from the disease (it is considered that only symptomatic individuals have a case fatality rate to be considered). Additionally, once recovered, immunity is not permanent, and an individual may become susceptible (S) to contracting the disease again. An intervention measure such as vaccination has also been considered, providing temporary immunity against the disease. This dynamic is represented by the flow diagram shown in Figure 1.

In the proposed model, vital dynamics are considered, meaning that there is a constant influx of people into the system (through birth or immigration) and that there is natural mortality (not due to the disease). Both assumptions are accounted for by the recruitment rate (Λ) and the natural death rate (μ), respectively. Regarding the dynamics of the disease, differentiated transmission rates are considered for symptomatic (β_1_) and asymptomatic (β_2_) individuals; the incubation rate (η) determines how an exposed individual becomes infectious, along with a probability of developing symptoms (p) or not (1 − p). Additionally, symptomatic individuals may be isolated at an isolation rate (φ). Infected individuals either die from the disease at a case fatality rate (ω) or recover at recovery rate (γ), which are similar for symptomatic and asymptomatic individuals, but differ from the recovery rate for isolated individuals (δ), who possibly experienced more severe symptoms and therefore have a longer recovery time (i.e., a lower recovery rate). With respect to immunity, two types are lost at natural immunity loss rate (σ) and acquired immunity loss rate (κ), respectively. Vaccination is assumed to confer protection against infection, with the degree of protection determined by vaccine effectiveness (ε). The vaccination rate (ν) measures the speed and extent of the immunization process, understood as the proportion of individuals vaccinated relative to the total susceptible population per unit of time (for example, if the health system can vaccinate 2000 people per day in a population of 100,000, the vaccination rate is 0.02, meaning 2% of the susceptible population is vaccinated per day). The effectiveness of vaccination (ε) is the percentage indicating the degree or level of effective protection the vaccine provides to the inoculated population. These dynamics are formalized mathematically as:(1)$$\left\{\begin{array}{c} \frac{dS}{dt}\;=\;\Lambda -\frac{\left(\beta_{1}I+\beta_{2}A\right)S}{N}-\varepsilon \nu S+\kappa V+\sigma R-\mu S,\;S\left(0\right)\;=\;S_{0}\\ \frac{dE}{dt}\;=\;\frac{\left(\beta_{1}I+\beta_{2}A\right)S}{N}-\eta E-\mu E,\;E\left(0\right)\;=\;E_{0}\;\\ \frac{dI}{dt}\;=\;p\eta E-\varphi I-\gamma I-\omega I-\mu I,\;I\left(0\right)\;=\;I_{0}\;\\ \frac{dA}{dt}\;=\;\left(1-p\right)\eta E-\gamma A-\mu A,\;A\left(0\right)\;=\;A_{0}\;\\ \frac{dQ}{dt}\;=\;\varphi I-\delta Q-\omega Q-\mu Q,\;Q\left(0\right)\;=\;Q_{0}\\ \frac{dR}{dt}\;=\;\gamma I+\gamma A+\delta Q-\sigma R-\mu R,\;R\left(0\right)\;=\;R_{0}\\ \frac{dV}{dt}\;=\;\varepsilon \nu S-\kappa V-\mu V,\;V\left(0\right)\;=\;V_{0}\;\\ \theta \;=\;(\Lambda ,\beta_{1},\beta_{2},\;\varepsilon ,\;\nu ,\kappa ,\sigma ,\mu ,\eta ,p,\varphi ,\gamma ,\delta ,\omega )\in R_{+}^{14} \end{array}\right.$$

The model is represented by a system of ordinary differential equations with parameters denoted by vector θ, with 0 ≤ p ≤ 1, 0 ≤ ε ≤ 1. All initial conditions are non-negative. Additionally, the total population can be found as:(2)$$N\left(t\right)\;=\;S\left(t\right)+E\left(t\right)+I\left(t\right)+A\left(t\right)+Q\left(t\right)+R\left(t\right)+V\left(t\right).$$

Since the model applies to human populations, solutions must be non-negative, ensuring they remain biologically feasible.

### 2.2. Simulations and Sensitivity Analysis

This research employed Python 3.10.12 on Google Colaboratory for simulations, parameter estimation, and sensitivity analysis. NumPy v1.26.4 [23] was used for scientific computations. SciPy v1.12.0 [24] provided numerical methods, including differential equation solving (*integrate.odeint*) and normal probability distribution assignment (*stats.norm*) for PRCC calculations. SALib v1.4.8 [25,26] facilitated model sensitivity analysis using its Sobol module. Matplotlib v3.8.3 [27] generated and customized all visualizations.

#### 2.2.1. Simulations

Parameter ranges for our model were derived from official reports and studies associated with the 2022 monkeypox outbreak [17,22,28,29,30,31,32,33,34,35,36,37,38] (details in Appendix A). These values are presented in Table 1 and were used to explore two epidemiological scenarios: disease-free and endemic.

#### 2.2.2. Sensitivity Analysis

Sensitivity analysis evaluates how changes in model parameters influence outputs. In epidemiological models, it helps us understand how variations in birth rates, transmission rates, and other parameters, affect disease dynamics (susceptible, infected, recovered populations) and the basic reproduction number (R_0_). We assign probability distributions to parameters and random sample values. Correlation indices [39] capture both linear and non-linear effects on model variables.

A two-part sensitivity analysis was conducted. First, we assessed the impact of each parameter on R_0_, identifying key parameters for controlling initial epidemics through vaccination. Second, Sobol sensitivity indices quantified the sensitivity of dynamic behavior (population sizes over time) to parameter variations [40,41,42,43]. (details in Appendix A).

### 2.3. Case Study of the Epidemiological Mpox Outbreak of 2022

A comparative approach assessed vaccination’s impact on global and continental populations (Europe, South America, North America). We simulated model dynamics with and without vaccination. Unvaccinated simulations established baseline parameters (i.e., with κ = 0, ε = 0 and ν = 0). Vaccinated simulations explored the effect of varying vaccination parameters. Graphical results presented model sensitivity to individual vaccination parameters.

#### 2.3.1. Parameter Estimation and Model Fitting

Parameter estimation for the non-vaccination model utilized Python’s Pandas and lmfit v1.2.2 [44]. The L-BFGS-B algorithm within *lmfit.minimize* function facilitated optimization. This quasi-Newton method, a variant of the Broyden–Fletcher–Goldfarb–Shanno (BFGS) method [45], tackles high-dimensional, non-linear problems. It allows for specifying parameter variation limits during the minimization process [45,46]. The objective function minimized was the cost function J(θ), defined as:$$J\left(\theta \right)=\;\sum_{i=1}^{N}{\left(y_{i}(\theta )-{\hat{y}}_{i}\right)}^{2}$$
where θ is the parameter vector of the model, N is the number of points in the actual data, $y_{i}(\theta )$ is the cumulative symptomatic infected value of the model in the time $t_{i}$ for the set of parameters θ, and ${\hat{y}}_{i}$ is the accumulated value of real data in time $t_{i}$. The goal is to find the θ parameters that minimize this cost function, i.e.:$$\hat{\theta }\;=\;\underset{\theta }{\mathrm{arg min}}J(\theta )$$where $\hat{\theta }$ are the optimized parameters that best fit the SEIR model to the actual accumulated data.

In our model, I(t) denotes the actual number of symptomatically infected individuals, while surveillance data corresponds to reported/diagnosed cases. Thus, observed cumulative cases do not exactly equal true symptomatic cases. To account for this, we introduce a reporting factor ρ∈(0,1] that captures underreporting, so that the observed symptomatic cases are given by:$$I_{obs}\left(t\right)=\rho \cdot I(t)$$
where $I_{\mathrm{obs}\;}\left(t\right)$ denotes the observed symptomatic cases. Accordingly, the cumulative function $y\left(t\right)$ fitted to the data is computed as:$$y\left(t\right)=\int_{0}^{t}\;\rho \cdot I(s)ds$$

This formulation clarifies that fitting directly with I(t) (assuming ρ=1) is a simplification, while incorporating ρ provides a more realistic mapping between the model and surveillance data [47,48,49].

The data used for model fitting was obtained from Our World in Data—Mpox Data Explorer [4]. It included daily and accumulated cases, categorized globally and by continent (Europe, South America, North America) for the periods 1 May 2022 to 30 April 2023 (Appendix A).

For model adjustment and parameter estimation, we utilized the accumulated infected cases. Initial conditions for symptomatic infected individuals I(0) were derived from the tabulated data. Considering that only susceptible S(0) and symptomatic infected I(0) populations existed at the outbreak’s beginning, the initial condition vector can be written as (S(0),E(0),I(0),A(0),Q(0),R(0),V(0) = (S(0),0,I(0),0,0,0)). The ranges of initial values and conditions assumed for each study case are detailed in Appendix A.

Likewise, the adjustment level was quantified using the correlation coefficient (R^2^), the Akaike Information Criterion (A.I.C.) and the Bayesian Information Criterion (B.I.C.).

#### 2.3.2. Potential Impact of Vaccination

To analyze the impact of vaccination and isolation on the epidemic’s progression according to the model’s dynamics, sensitivity plots were generated for these interventions’ parameters.

## 3. Results

### 3.1. Model Building

#### 3.1.1. Positivity

**Theorem** **1.***Let the ordinary differential equation system (1) of the Mpox disease dynamics be given with initial conditions* $S\left(0\right)\geq 0,E\left(0\right)\geq 0,I\left(0\right)\geq 0,A(0)\geq 0,Q(0)\geq 0,R(0)\geq 0,V(0)\geq 0$*, then, system (1) has nonnegative solutions* $(S,E,I,A,Q,R,V)\in R_{+}^{7}$* for all time* t>0*. (Proof in Appendix A)*

#### 3.1.2. Boundedness

**Theorem** **2.**
*The invariant feasible region of the system (1) for Mpox disease, is a non-negative region defined by*


$$\Omega =\left\{\left(S\left(t\right),E\left(t\right),I\left(t\right),A\left(t\right),Q\left(t\right),R\left(t\right),V\left(t\right)\right)\in R_{+}^{7}:N\left(t\right)\leq \frac{\Lambda }{\mu }\right\}$$*where *$N\left(t\right)\;=\;S\left(t\right)+E\left(t\right)+I\left(t\right)+A\left(t\right)+Q\left(t\right)+R\left(t\right)+V\left(t\right)$*, and *Ω* attracts all the solutions in *$R_{+}^{7}$*. (Proof in Appendix A)*.

#### 3.1.3. Existence of the Equilibrium Points

To determine the equilibrium points, which are the constant solutions of the system (1), there should be no changes in each of the populations. Mathematically this represents that we must set each of the equations equal to zero.

##### Disease-Free Equilibrium Point

**Theorem** **3.**
*System (1) always has a unique disease-free equilibrium point. (Proof in Appendix A).*


##### Estimation of the Basic Reproduction Number (R_0_)

The next-generation matrix method [50,51] was used to calculate the basic reproduction number (R_0_). (Estimation of R_0_ in Appendix A)$$R_{0}=\frac{(\kappa +\mu )\eta }{\left(\eta +\mu )(\varepsilon \nu +\kappa +\mu \right)}\left[\frac{p\beta_{1}}{\left(\varphi +\gamma +\omega +\mu \right)}+\frac{\left(1-p\right)\beta_{2}}{\left(\gamma +\mu \right)}\right]$$

##### Endemic Equilibrium Point

**Theorem** **4.**
*The system (1) admits a unique endemic equilibrium *

$P^{*}$

* with *

$I^{*}>0$

* if and only if *

$R_{0}>1$

*. (Proof in Appendix A).*


#### 3.1.4. Local Stability Analysis

To study the stability of the nonlinear system (1), we calculated the Jacobian matrix. We evaluated this matrix at an arbitrary equilibrium point P = (S,E,I,A,Q,R,V), denoted by:$$J\left(P\right)\;=\left(\begin{array}{ccccccc} -\frac{\left(\beta_{1}I+\beta_{2}A\right)}{N}-\varepsilon \nu -\mu & 0 & -\beta_{1}\frac{S}{N} & -\beta_{2}\frac{S}{N} & 0 & \sigma & \kappa \\ \frac{\beta_{1}I+\beta_{2}A}{N} & -(\eta +\mu ) & \beta_{1}\frac{S}{N} & \beta_{1}\frac{S}{N} & 0 & 0 & 0\\ 0 & p\eta & -(\varphi +\gamma +\omega +\mu ) & 0 & 0 & 0 & 0\\ 0 & (1-p)\eta & 0 & -(\gamma +\mu ) & 0 & 0 & 0\\ 0 & 0 & \varphi & 0 & -(\delta +\omega +\mu ) & 0 & 0\\ 0 & 0 & \gamma & \gamma & \delta & -(\sigma +\mu ) & 0\\ \varepsilon \nu & 0 & 0 & 0 & 0 & 0 & -(\kappa +\mu ) \end{array}\right)$$

##### Disease-Free Equilibrium Point

**Theorem** **5.**
*The endemic equilibrium point is locally asymptotically stable if and only if both R_0_ > 1 and conditions (25) are satisfied. (Conditions and proof in Appendix A).*


##### Endemic Equilibrium Point

**Theorem** **6.**
*The disease-free equilibrium point is locally asymptotically stable if and only if R_0_ < 1. Otherwise, it is unstable [52]. (Proof in Appendix A)*


### 3.2. Simulations and Sensitivity Analysis of the Model

#### 3.2.1. Simulations

Results of the epidemiological scenarios are presented under a disease-free scenario (Figure 2) and under endemic scenario (Figure 3). Specific values, indicated by the gray box in the figures, were chosen to represent these scenarios. We also used the initial conditions: S(0) = 599,999, I(0) = 1, and the rest of populations equal to zero (we varied the values of β_1_, β_2_ and p to illustrate these cases). In the disease-free scenario, the four infected populations gradually disappear after an initial outbreak (Figure 2). This is confirmed by the phase plot (Appendix A) showing the trajectory between susceptible and infected individuals. Even after simulating for a long period (up to 60,000 days), the populations approach zero, indicating that the disease-free equilibrium is an attractive state. In the endemic case, the infected populations fluctuate over time and do not disappear (Figure 3). This is evident in the phase plot (Appendix A), which shows the infected populations persisting even after simulating 60,000 days. Additionally, the phase plot demonstrates that the equilibrium point in this scenario is also attractive.

#### 3.2.2. Sensitivity Analysis of the Model

##### Sensitivity Analysis for R_0_

Figure 4 presents a tornado plot summarizing the sensitivity analysis of R_0_. We used Latin hypercube sampling (LHS) with 1024 samples and a random seed of 1221 in Python. Based on the values of means for each parameter of the model (Table 1), most of them followed a normal probability distribution except by the probability of developing symptoms (p) and the case fatality rate (ω), which had uniform distributions (Appendix A). Partial rank correlation coefficients (PRCC) were calculated, and bootstrap analysis with 1000 samples was performed to estimate PRCC confidence intervals (Appendix A).

The sensitivity analysis revealed significant negative correlations for parameters: recovery rate (γ) (PRCC index absolute value: 0.5102), vaccination rate (ν) (PRCC index absolute value: 0.4643), and effectiveness of vaccination (ε) (PRCC index absolute value: 0.1048). Conversely, it showed positive correlations for: transmission rate for symptomatic (β_1_) (PRCC index absolute value: 0.3181), transmission rate for asymptomatic (β_2_) (PRCC index absolute value: 0.2671) and acquired immunity loss rate (κ) (PRCC index absolute value: 0.1883). These results indicate that: (1) increasing the recovery rate, the vaccination rate, and the effectiveness of vaccination, and/or (2) decreasing the transmission rates and the acquired immunity loss rate; all contribute to reducing the basic reproductive number R_0_ and controlling the epidemic outbreak.

Based on this sensitivity analysis, we constructed a decision surface for R_0_ with respect to vaccination rate (ν) and effectiveness of vaccination (ε) (Figure 5). The plane representing R_0_ = 1 (purple plane in Figure 5A, red line in Figure 5B) separates the disease-free and endemic scenarios. We observe that combined strategies that increase vaccination speed and improve vaccine efficiency can control the epidemic, avoid the endemic state and maintain R_0_ < 1, (large blue region in Figure 5A,B). For example, a high vaccination rate ν > 0.02 and vaccine effectiveness ε > 0.7 can keep the epidemic disease-free (Figure 5B).

##### Sensitivity Analysis for the Model

We analyzed how parameter variations affect the behavior of the symptomatic infected population (I) using Sobol sensitivity indices (Figure 6 and Figure 7). To perform this analysis, we used the *sobol.sample* function in Python to generate 3,932,160 random samples (with $N=2^{16}=131072,D=14$ parameters). The number of samples was chosen based on N(2D+2), because it is computed when we calculated S2 [25,26]. We used a random seed of 1221, and the model was simulated for 730 days with initial populations of S(0)=599999,I(0)=1 and the rest of populations equal to zero. The range of the values of the analyzed parameters are specified in Table 1. The analysis varied all parameters across their entire ranges (input factors) and calculated the sensitivity indices with respect to the symptomatic infected population (output variable). Detailed sensitivity index values are provided in (Appendix A).

Figure 6A presents a tornado plot for all parameters, showing both first order (S1) and total sensitivity indices (ST). This allows us to visualize the relative contribution of each parameter’s individual effect (S1) to its overall impact on the variability (ST). In simpler terms, it highlights how much each parameter directly influences the symptomatic infected population, compared to its total influence accounting for interactions with other parameters. Figure 6B displays the S1 indices sorted by their overall sensitivity, revealing the most influential parameters. Both images show that the mathematical model is highly sensitive to variations in case fatality rate (ω), incubation rate (η), and to a lesser extent to the isolation rate (φ), vaccination rate (ν), natural immunity loss rate (σ), recruitment rate (Λ), the probability of developing symptoms (p) and the recovery rate of isolated (δ). Interestingly, while parameters ω and η exhibit the strongest individual effects on the infected (S1), their combined effects with other parameters (ST—S1) contribute most significantly to the overall variability (ST). Appendix A illustrates these main effects of parameters ω and η, respectively. Appendix A shows how increasing the case fatality rate affects the dynamics of infected patients. As the rate increases, the peak number of infected patients significantly decreases. In extreme values, the model transitions from an endemic state (R_0_ > 1) to a disease-free case (R_0_ < 1) where the infection eventually remains indetectable. In contrast, Appendix A demonstrates the effect of a higher incubation rate. Here, the peak number of infected patients increases significantly with a rising incubation rate. Additionally, the model remains in an endemic state throughout the simulation.

Figure 7 uses a heatmap to visualize the interaction effects between pairs of parameters on the variability of the symptomatic infected population. Positive values indicate a synergistic effect, where the combined influence of two parameters is greater than their individual effects. Conversely, values near zero suggest weak or negligible interaction between the parameters regarding their impact on the model’s output. Notably, the heatmap highlights that the interaction between the case fatality rate (ω) and the rate of loss of natural immunity (σ) has the strongest influence on the variability of the symptomatic infection curve. Appendix A exemplifies this joint effect. It shows that simultaneously increasing the case fatality rate ω and the rate of loss of natural immunity σ leads to a lower endemic equilibrium point (ω = 0.00008, σ = 0.0027) compared to the scenario where these parameters change in opposite directions (in Appendix A with ω = 0.00008, σ = 0.04), with decreasing σ, while the value of R_0_ is not affected.

### 3.3. Case Study of the Epidemiological Mpox Outbreak of 2022

#### 3.3.1. Parameter Estimation and Model Fitting

Table 2 summarizes the initial conditions for the model without vaccination, and the results of the optimal parameters fitted to the data. Additionally, Figure 8 visually depicts the model’s fit for each region.

An additional fitting was implemented. Since reported cases correspond to flows of transitions between compartments rather than the stock of individuals in a given compartment, this approach also considers the appropriate flow:If cases are interpreted as the onset of symptoms, the relevant flow is: 


$$\mathrm{Incidence}\;\left(t\right)=\rho p\eta E(t),$$
that is, the proportion ρ of exposed individuals who progress to the symptomatic infectious phase.


If, alternatively, the data corresponds to reported cases, the correct flow is: 


$$Reported\left(t\right)=\rho \phi I(t),$$
which reflects the detection and notification rate of symptomatic infectious individuals in the compartment I(t).

Accordingly, the model directly produces the daily modeled incidence, and cumulative cases are obtained by integrating this flow over time:$$C\left(t\right)=\int_{0}^{t}\;\;\mathrm{Incidence}\;(s)ds$$

This procedure emphasizes that observational data should be compared against incidence flows rather than the number of individuals in a compartment [53,54]. Other studies apply the same principle when fitting SEIR-type models to COVID-19 data [55,56].

Table 3 presents the adjustment metrics between the model and the data for each geographic region.

#### 3.3.2. Potential Impact of Vaccination

The potential impact of vaccination-associated parameters reflects notorious changes in the symptomatic infection curve for each geographic region (Figure 9 and Appendix A). Figure 9A demonstrates the significant effect of vaccination effectiveness in reducing the peak number of infected individuals in the global population. Thus, the peak decreases from 3863 (without vaccination) to 3210 with the least efficient vaccine. Additionally, higher vaccine efficiency reduces both the peak number of infected individuals (I_max_) and the basic reproduction number R_0_. In the depicted endemic scenarios, all curves trend towards a similar equilibrium value. This epidemiological behavior is also observed at the level of individual continental populations (Appendix A). Similarly to increasing efficiency, a higher vaccination rate reduces the peak number of infected individuals (I_max_) and R_0_. This leads all scenarios to reach an endemic state at both the world level (Figure 9B) and within individual continental populations (Appendix A). On the other hand, Figure 9C demonstrates the benefit of prolonged vaccine protection at the world level. Decreasing the rate of acquired immunity loss (extending the effective protection period) reduces both the peak infected population (I_max_) and R_0_, leading to a more favorable epidemic prognosis. This scenario is also observed at the level of individual continental populations (Appendix A). Finally, Figure 9D explores the combined effect of isolation rate and vaccination strategy (using average vaccination parameters from Table 1) for the world population. Increasing the isolation rate (decreasing patient detection time) consistently lowers the peak infection across all regions. Again, this scenario is also observed at the level of individual continental populations (Appendix A).

## 4. Discussion

Diverse mathematical approaches have been used to study the epidemiology and transmission of Mpox [22]. These include regression [57,58], compartmental [17,59], branching process [13,15], stochastic Monte Carlo [60], agent-based (ABMs) [61], and network [62] models. Among them, compartmental models are among the most commonly used. This study addresses the composition of a compartmental mathematical model to study the dynamics of Mpox. The implementation is based on an extended SEIR model, allowing for the addition of highly epidemiologically relevant subpopulations, such as asymptomatic, isolated (quarantined), and vaccinated individuals, together with dynamic rates in each of them. This model was subsequently applied to continental-scale data, in contrast to previous studies that either considered these characteristics separately or focused on animals and employed mathematical tools such as fractional differential equations [16,35,63,64]. In addition, our approach has the advantage of a direct biological interpretation of exchange rates compared to those derived from fractional-order models. Beyond these methodological aspects, our study specifically addresses the scientific gap of quantifying how vaccination and isolation interact under conditions of asymptomatic transmission and waning immunity—factors often underreported yet epidemiologically relevant in Mpox. By explicitly modeling vaccinated but still infectious individuals, incorporating immunity decay, and validating the model with continental-scale real-world data, we provide novel insights into how different parameters distinctly affect the basic reproduction number (R_0_) versus the overall epidemic burden.

Asymptomatic transmission has emerged as a critical driver of silent epidemic spread and is a key component of our model, which differentiates between symptomatic and asymptomatic individuals. Despite lacking clinical signs, asymptomatic infections contribute meaningfully to overall transmission, a finding supported by modeling studies and time-series analyses [65,66]. Recent reviews have underscored the underreporting of Mpox cases, urging consideration of asymptomatic infections as potential sources of transmission [11,67,68]. Evidence also suggests that vaccinated individuals may act as asymptomatic carriers, further reinforcing the need to include this group in contact tracing and public health strategies [28,69]. Although the exact infectivity of asymptomatic cases has not been clearly defined, our model assigns a differentiated transmission rate to reflect their potential impact on disease spread.

Epidemiologic models include isolation as a key intervention, in accordance with current public health guidelines [16]. Integrating isolation and quarantine protocols for symptomatic individuals remains essential to curbing Mpox transmission, particularly in the early stages of an outbreak. Behavioral change, as a way of modulating isolation, has been recognized as a critical determinant in shaping the course of the Mpox outbreak. Multiple modeling studies have shown that reductions in high-risk sexual behavior among men who have sex with men (MSM) led to early and significant decreases in transmission—often preceding the widespread effects of vaccination campaigns [13,70]. Our findings support this, indicating that even modest declines in contact rates can delay epidemic peaks or prevent extensive outbreaks [71]. However, beyond behavioral adaptations, the timely isolation of symptomatic individuals stands out as one of the most effective standalone interventions. Simulation models have demonstrated that early and rapid isolation, particularly when combined with vaccination strategies, can reduce total infections and control disease transmission [36,72,73]. This intervention is especially impactful when implemented before community-wide transmission becomes established, underscoring the importance of early outbreak detection and a rapid public health response [17,74]. Our model considers previous research indicating that waning immunity—whether resulting from past infection or historic smallpox vaccination—plays a critical role in the potential resurgence of Mpox, particularly in the absence of timely booster strategies [14,75]. This issue is especially pertinent among older individuals vaccinated decades ago, whose residual protection may now be insufficient [76,77]. As such, mathematical epidemiological models must account for both vaccination status and immunity decay over time to more accurately predict outbreak dynamics and guide public health interventions. Integrating these factors is essential for evaluating long-term control strategies, assessing population vulnerability, and designing effective immunization policies in the context of re-emerging infectious diseases like Mpox.

Our simulation and qualitative analysis demonstrate that the proposed model captures two distinct epidemiological scenarios, determined by the basic reproduction number R_0_. When R_0_ < 1, the system converges to a disease-free equilibrium, while for R_0_ > 1, an endemic state may emerge. In the endemic regime, symptomatic infections may eventually decline, yet a portion of the population remains susceptible. Moreover, the model reveals that, depending on parameter values, the system can exhibit multiple infection peaks over time, suggesting the potential for recurrent waves of transmission.

The sensitivity analysis revealed notable differences between the parameters that most influence the long-term behavior of the infected population and those that predominantly affect the basic reproduction number, R_0_. Specifically, the model outcomes related to the number of infected individuals were highly sensitive to changes in the parameters case fatality rate (ω) and incubation rate (η), and to a lesser extent to the isolation rate (φ), vaccination rate (ν), natural immunity loss rate (σ), recruitment rate (Λ), the probability of developing symptoms (p) and the recovery rate of isolated (δ). In contrast, the parameters with the greatest influence on R_0_ were recovery rate (γ), vaccination rate (ν), effectiveness of vaccination (ε), transmission rate for symptomatic (β_1_), transmission rate for asymptomatic (β_2_) and acquired immunity loss rate (κ). This distinction has important implications. From a theoretical perspective, it suggests that achieving disease control can be approached in multiple ways. Modifying parameters that strongly influence R_0_ may allow us to reduce this threshold below 1, theoretically leading to disease elimination. This includes strategies such as increasing the recovery rate, the vaccination rate, and the effectiveness of vaccination, or reducing the transmission rates and the acquired immunity loss rate. However, the analysis also highlights that lowering R_0_ does not necessarily translate into a proportionate reduction in epidemic severity, especially in terms of peak prevalence or long-term number of infected individuals, as discussed in a previous study for COVID-19 [78]. For instance, the incubation rate (η) had minimal impact on R_0_, yet variations in this parameter produced dramatic changes in the infection dynamics. A reduction in η led to a significant decrease in the peak number of infections, from approximately 30,000 to fewer than 5000, despite R_0_ remaining nearly constant and the system staying within an endemic regime. This finding underscores the importance of considering both the reproduction number and the temporal dynamics of infection when designing public health interventions [78]. Even if the disease cannot be eradicated (i.e., R_0_ > 1), reducing the peak burden on the healthcare system can be achieved through targeted modifications of parameters such as the incubation period. This insight is valuable for epidemic management, as it supports strategies that mitigate healthcare system overload and improve overall public health outcomes, even in scenarios where disease elimination is not immediately attainable.

The case fatality rate (CFR, ω) emerged as a parameter with a pronounced influence on both the overall dynamics of the infected population and the basic reproduction number, R_0_. An increase in this rate not only leads to a substantial reduction in the peak number of infections and a temporal delay in the epidemic peak but can also shift the system from an endemic to a disease-free equilibrium. This dual influence positions the case fatality rate, alone or in combination with other sensitive parameters, as a critical determinant in shaping both short-term epidemic behavior and long-term disease outcomes. In some scenarios, an increase in CFR reduces R_0_ below 1, resulting in disease elimination, while simultaneously decreasing the maximum number of infected individuals. Furthermore, interactions between the CFR and other parameters, such as the natural immunity loss rate (σ), significantly modulate disease dynamics. It was noted that the lowest values of these rates can initially lead to higher infection peaks but ultimately converge to a lower endemic equilibrium. Other parameter combinations may drive the system toward disease elimination altogether [79]. In most infectious diseases, CFR is determined by the intrinsic pathogenicity of the causative agent—shaped by its genetic composition and its capacity to disrupt host physiological balance. The lethality of viruses such as HIV or smallpox is a function of their virulence, which often inversely correlates with transmissibility [80]. More virulent strains tend to spread less efficiently due to host incapacitation or death, thus limiting widespread transmission. Additionally, virulence is not static; it may evolve over time through host–pathogen interactions, influenced by factors such as transmission route, population density, host immunity, and the broader ecological context [81]. Consequently, although CFR has a demonstrable impact on epidemic dynamics within the model, it remains a parameter that is not readily subject to direct intervention. Nonetheless, understanding its role is essential for anticipating potential epidemic outcomes and identifying complementary strategies—such as vaccination, mitigation of transmission, or healthcare interventions—that can indirectly reduce disease burden even in the absence of direct control over lethality.

Although our analysis focuses on the 2022–2023 outbreaks in Europe and the Americas, which were largely sustained by human-to-human transmission [3], it is essential to acknowledge that Mpox remains endemic in several African regions, where recurrent zoonotic spillovers and a continuous human burden are significant. This highlights the importance of studying both endemic and non-endemic settings, and while our model is centered on the human system, the framework could be extended to include animal reservoirs when appropriate data are available. Therefore, the model developed to retrospectively analyze the epidemic dynamics in Europe and the Americas and to evaluate the impact of vaccination strategies, demonstrated a good fit to the observed data on cumulative symptomatic infections. This suggests that, within the framework and assumptions of the model, it captures key features of the outbreak dynamics and has explanatory potential regarding the temporal progression of symptomatic cases. Based on this foundation, we explored the effect of different vaccination-related parameters on epidemic control. Among these, the vaccination rate emerged as the most influential factor in reducing the peak number of infections. This observation is consistent with the sensitivity analysis, which identified the vaccination rate parameter (ν) as having a greater impact on model outcomes compared to other vaccination-related parameters. Furthermore, increasing the vaccination rate consistently led to a significant reduction in the basic reproduction number, R_0_, reinforcing its central role in shaping long-term epidemic trajectories [33]. Other interventions, such as increasing the isolation rate (φ), reducing the acquired immunity loss rate (κ), and enhancing the effectiveness of vaccination (ε), also contributed to reducing and delaying the infection peak, albeit with a more moderate impact. These results highlight that while several vaccination-related strategies can influence epidemic behavior, the rapidity of vaccine administration stands out as the most effective measure for controlling both the magnitude and timing of the outbreak [82]. In summary, the vaccination rate not only influences the immediate dynamics of symptomatic infection but also plays a critical role in determining whether the epidemic can be driven below the threshold for sustained transmission. This underscores the importance of rapid and widespread vaccination campaigns as a primary tool in epidemic management, particularly during the early phases of an outbreak when reducing peak incidence and achieving long-term control are most urgent. From a public health perspective, our model can thus serve as a decision-support tool by highlighting which parameters most strongly influence epidemic dynamics. In practice, this allows policymakers to prioritize rapid vaccination campaigns, strengthen case isolation strategies, and design booster schedules to counteract waning immunity. Furthermore, the model’s ability to distinguish between drivers of R_0_ and determinants of epidemic burden provides actionable insights for allocating resources and mitigating healthcare system overload. At the same time, it is important to acknowledge both the advantages and limitations of mathematical models in infectious disease research. Their strengths include simulating outbreak scenarios, quantifying the influence of epidemiological parameters, and anticipating the effects of interventions before implementation. However, they also rely on assumptions and parameter accuracy, and may not fully capture behavioral dynamics, ecological drivers, or stochastic events. Recognizing these pros and cons is essential for interpreting our results as complementary evidence to epidemiological and clinical data.

Our modeling framework inevitably involves some simplifying assumptions, particularly in the fitting procedure. The compartment I(t) represents the true number of symptomatically infected individuals, while available surveillance data consists of reported or diagnosed cases, which are generally fewer. To reconcile this gap, we introduced a constant reporting factor ρ∈(0,1], such that the observed cases are given by $I_{\mathrm{obs}\;}(t)=\rho \cdot I(t)$, an approach widely used in epidemic modeling to account for underreporting [47,48,49]. This method provides a practical link between the model and the data without altering theoretical results such as the basic reproduction number or local stability analysis, though it remains a simplification. More detailed alternatives, such as incorporating an explicit compartment for diagnosed individuals, could further distinguish epidemiological dynamics from surveillance data [49], and represent a promising direction for future research, especially when underreporting is substantial or time dependent. In addition, our model currently focuses on the human system without explicitly incorporating the animal subsystem, a reasonable choice for the 2022–2023 outbreaks in non-endemic regions, which were mainly sustained by human-to-human transmission, but less suited to endemic contexts where wildlife reservoirs and recurrent spillovers are essential for Mpox persistence. Extensions of the framework remain compatible with the inclusion of animal compartments or coupled human–animal subsystems, thereby enabling the assessment of cross-species transmission risks and long-term control strategies in settings where the zoonotic component cannot be overlooked.

## 5. Conclusions

The mathematical model for Mpox developed in this study incorporates asymptomatic individuals and two key public health interventions: isolation and vaccination. The parameter with the greatest influence on both the dynamics of symptomatic infections and the basic reproduction number R_0_ is the case fatality rate (ω). However, controlling this parameter directly is impractical. Among the parameters related to public health interventions, the vaccination rate (ν) has the most significant impact. Increasing the vaccination rate not only reduces R_0_ but also substantially lowers the peak number of infections.

The model suggests that transitioning from a potential endemic state to a disease-free state is achievable through a combination of control measures, such as increasing the vaccination rate (ν), improving vaccine effectiveness (ε), enhancing isolation rate (φ) and reducing immunity loss among vaccinated individuals (κ). These strategies contribute to lowering R_0_, reducing the maximum number of infections, and improving the long-term epidemic outlook by decreasing the total number of cases.

## Figures and Tables

**Figure 1 medsci-13-00226-f001:**
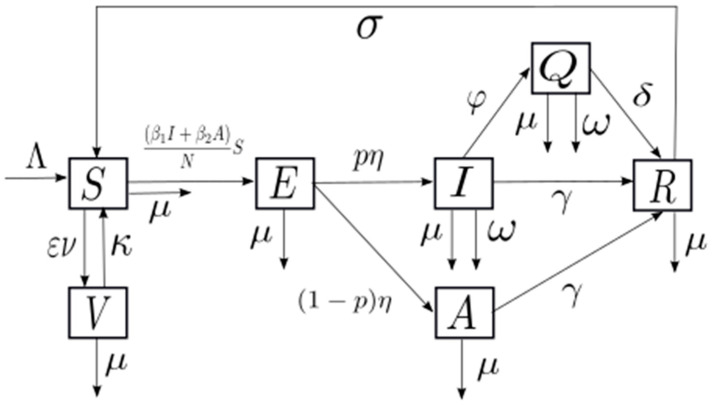
A flow diagram of monkeypox disease transmission considering the mathematical model.

**Figure 2 medsci-13-00226-f002:**
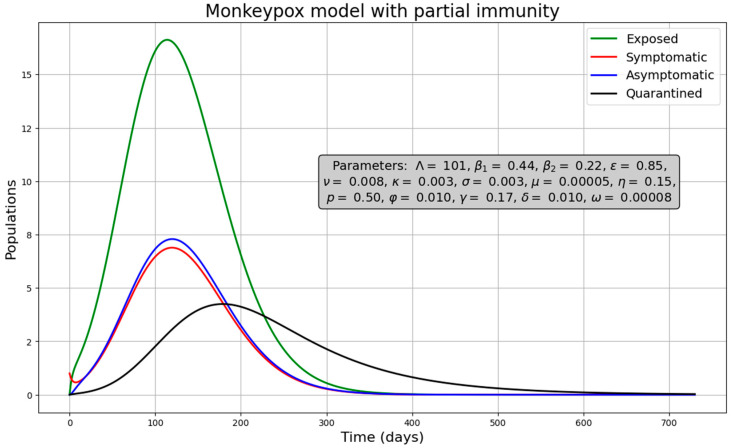
Solution curves under a disease-free scenario.

**Figure 3 medsci-13-00226-f003:**
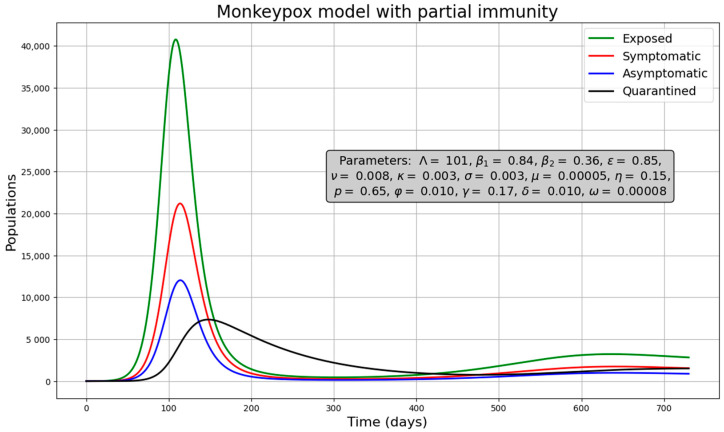
Solution curves under endemic scenario.

**Figure 4 medsci-13-00226-f004:**
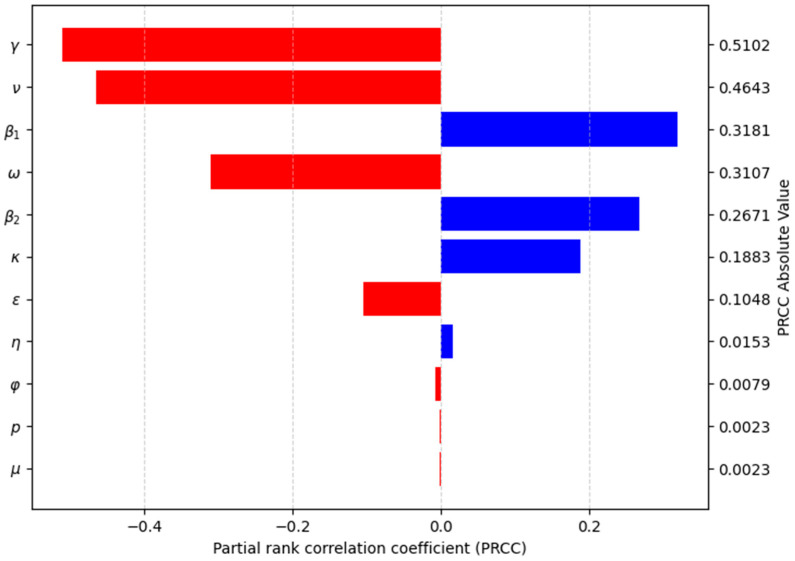
Tornado plot for sensitivity analysis of R_0_ regarding its parameters.

**Figure 5 medsci-13-00226-f005:**
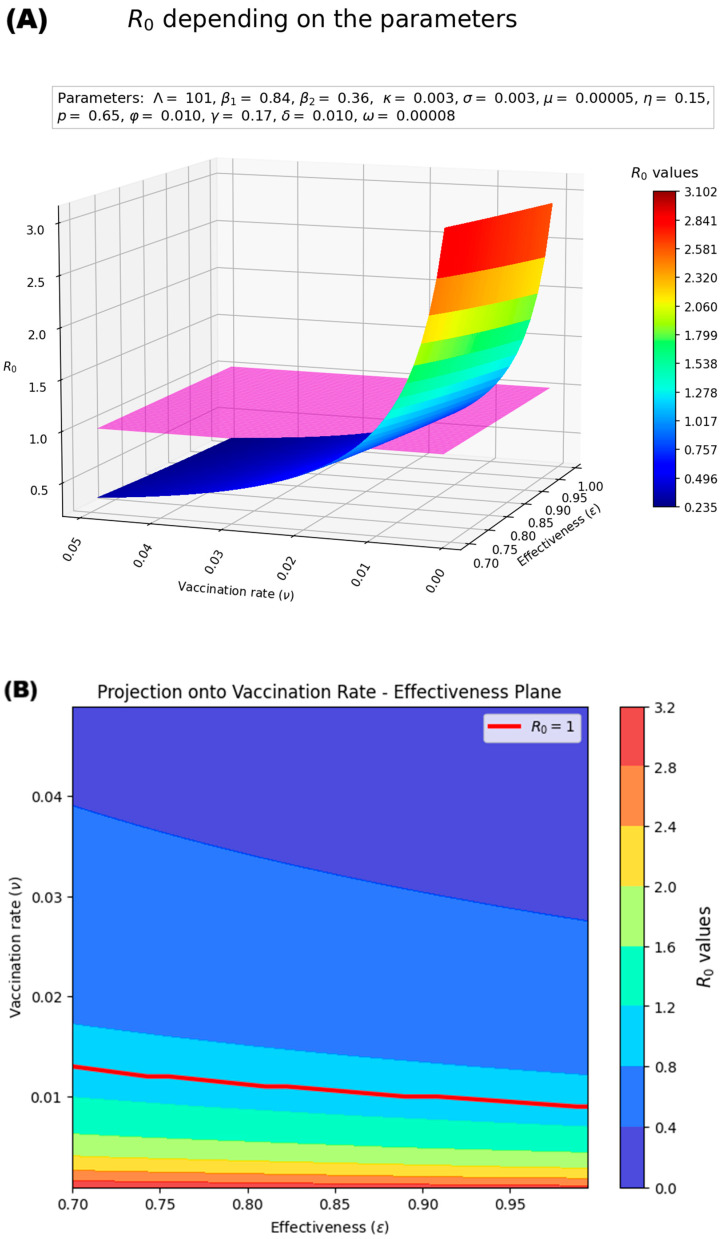
(**A**) Surface, and (**B**) Contour plot of R_0_ respect to variation in ε and ν.

**Figure 6 medsci-13-00226-f006:**
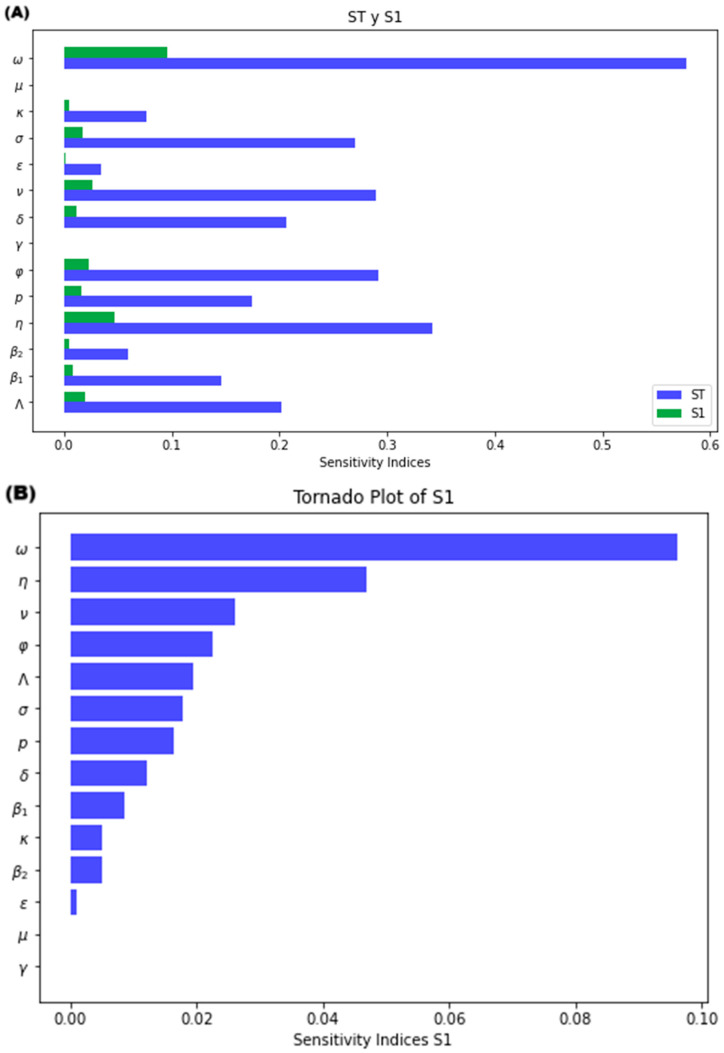
(**A**) Tornado plot for Sobol indices for S1 and ST, and (**B**) S1 descending order.

**Figure 7 medsci-13-00226-f007:**
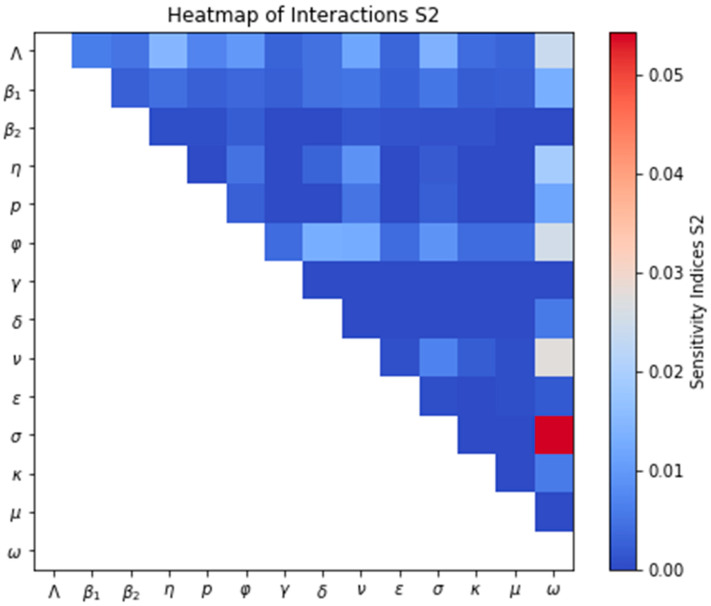
Heatmap for S2-Sobol sensitivity indices interaction.

**Figure 8 medsci-13-00226-f008:**
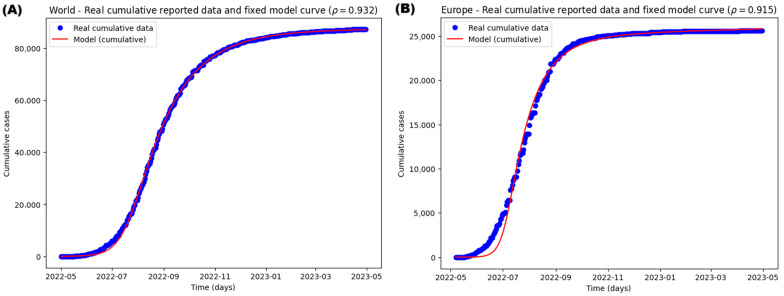
Parameter estimation and fitted curves of the actual and modeled data considering the accumulated data for the infected population shown for (**A**) World, (**B**) Europe, (**C**) South America, (**D**) North America.

**Figure 9 medsci-13-00226-f009:**
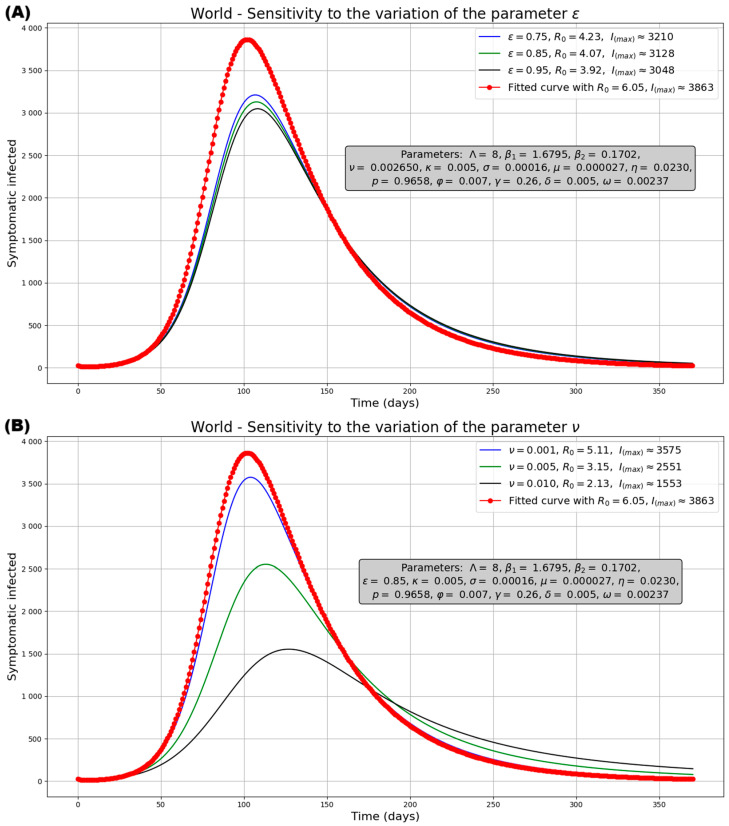
Effect of parameters variation on the infected population compared with the fitted curve without vaccination (with ε = 0, ν = 0, κ = 0) for the World. (**A**) Effect of vaccination effectiveness (ε) variation, (**B**) Effect of vaccination rate (ν) variation, (**C**) Effect of immunity lose rate (κ) variation, (**D**) Effect of isolation rate (φ) variation.

**Table 1 medsci-13-00226-t001:** Parameters of the model.

Parameter	Description	Range (Unit)	Reference
Λ	Recruitment rate	$\left[100.631507,\;3231.12082\right]person\times day^{-1}$	[30,31,32]
$\beta_{1}$	Transmission rate for symptomatic infected	$\left[0.4,\;1.0\right]\;day^{-1}$	[38]. Assumed
$\beta_{2}$	Transmission rate for asymptomatic infected	$\left[0.1,\;1.0\right]\;day^{-1}$	Assumed
η	Incubation rate	$\left[0.0476,\;0.2\right]\;day^{-1}$	[28,33,36,38]
p	Probability to develop symptoms	[0.4, 1]	Assumed
φ	Isolation rate	$\left[0.00001,\;0.01\right]\;day^{-1}$	Assumed
γ	Recovery rate for symptomatic and asymptomatic	$\left[0.0357,\;0.3333\right]\;day^{-1}$	[36,38]
δ	Recovery rate for isolated	$\left[0.00001,\;0.01\right]\;day^{-1}$	Assumed
ν	Vaccination rate	$\left[0.001,\;0.05\right]\;day^{-1}$	Assumed
ε	Effectiveness	[0.7, 1]	[36]
σ	Rate of loss of natural immunity	$\left[0.0027,\;0.04\right]\;day^{-1}$	[37]
κ	Rate of loss of immunity from vaccination	$\left[0.00317,\;0.0204\right]\;day^{-1}$	[37]
μ	Mortality rate	$\left[2.73973\times 10^{-6},\;6.02740\times 10^{-5}\right]\;day^{-1}$	[32]
ω	Lethality rate	$\left[0.00008,\;0.11\right]\;day^{-1}$	[29,34,35]

**Table 2 medsci-13-00226-t002:** Fitted parameters and initial conditions for the model without vaccination.

Parameter Estimation		Geographic Region			
Parameter or Initial Condition	Range	World	Europe	South America	North America
		Fitted Value (Initial Guess)	Fitted Value (Initial Guess)	Fitted Value (Initial Guess)	Fitted Value (Initial Guess)
S(0)	$\left[1,\;2\times 10^{6}\right]$	94,854.1383 (160,000)	26,132.5978 (25,000)	16,330.4531 (25,000)	103,602.999 (25,000)
I(0)	-	27 (27)	1 (1)	2 (2)	103 (103)
Λ	[1, 3231]	7.73970439 (34)	22.6202113 (30)	54.2253827 (30)	189.385924 (15)
$\beta_{1}$	[0.04, 2.5]	1.67953368 (1.2)	2.04177890 (1.2)	1.76850088 (1.2)	0.91634522 (0.9)
$\beta_{2}$	[0.01, 2.5]	0.17020184 (0.8)	0.39950298 (0.8)	0.37955379 (0.8)	0.66563355 (0.8)
η	[0.01, 0.2]	0.02295726 (0.08)	0.04005441 (0.08)	0.04000883 (0.08)	0.04191390 (0.05)
p	[0.1, 1]	0.96584613 (0.46)	0.94874630 (0.46)	0.94346047 (0.46)	0.44090917 (0.46)
φ	[0.00001, 0.09]	0.00717407 (0.006)	0.04094240 (0.03)	0.04788072 (0.03)	0.00466125 (0.006)
γ	[0.0357, 0.3333]	0.25943508 (0.06)	0.21399404 (0.06)	0.33142887 (0.06)	0.14785275 (0.06)
δ	[0.00001, 0.05]	0.00494816 (0.005)	0.00469751 (0.005)	0.00455631 (0.005)	0.00494974 (0.005)
σ	[0.0001, 0.04]	$1.6023\;\times 10^{-4}$ (0.003)	$1.0153\;\times 10^{-4}$ (0.003)	$1.0003\;\times 10^{-4}$ (0.003)	$3.6366\times 10^{-4}$ (0.003)
μ	$\left[1.0\times 10^{-6},\;9.0\times 10^{-4}\right]$	$2.7312\;\times 10^{-5}$ (0.00002)	$4.3514\;\times 10^{-6}$ (0.000005)	$1.6049\;\times 10^{-5}$ (0.000005)	$5.5609\;\times 10^{-6}$$\;(5\times 10^{-6}$)
ω	$\left[1\times 10^{-7},\;0.3\right]$	0.00237251 (0.00004)	$1.4828\;\times 10^{-5}$ (0.00004)	$2.7038\times 10^{-6}\;(0.00004)$	$9.2659\;\times 10^{-4}$$\;(4\times 10^{-5}$)
ρ	${\left[0.1,\;0.95\right]\;}^{*}$	0.93219423 (0.7)	0.91519545 (0.6)	0.94531585 (0.6)	0.61892306 (0.6)

* Although ρ can theoretically take values from 0 to 1, we assume that at least 5% of infected cases remain unreported.

**Table 3 medsci-13-00226-t003:** Adjustment values for models and actual data for each region studied.

Geographic Region	R^2^	A.I.C.	B.I.C.
World	0.9996	4824.20197	4894.40013
Europe	0.9952	4666.83938	4736.73918
South America	0.9973	4118.41144	4187.01198
North America	0.9989	4110.58893	4179.18947

## Data Availability

The data used for model fitting was obtained from Our World in Data—Mpox Data Explorer [4]. It included daily and accumulated cases, categorized globally and by continent (Europe, South America, North America) for the periods 1 May 2022 to 30 April 2023 (Appendix A).

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
