# Peer review of "A Compartmental Mathematical Model to Assess the Impact of Vaccination, Isolation, and Key Epidemiological Parameters on Mpox Control"

_medsci, 2025, doi:10.3390/medsci13040226_

Round 1

Reviewer 1 Report

Comments and Suggestions for Authors

Research article titled, “A Compartmental Mathematical Model to Assess the Impact of 2 Vaccination, Isolation, and Key Epidemiological Parameters on 3 Mpox Control” is an appreciable effort to understand and assess various parameters that could lead or contribute to disease-free state.

As stated by authors, the model suggests that transitioning from a potential endemic state to a disease-free state is achievable through a combination of control measures, such as increasing the vac- cination rate (ν), improving vaccine effectiveness (ε), enhancing isolation rate (φ) and re- ducing immunity loss among vaccinated individuals (κ). These strategies contribute to lowering Râ‚€, reducing the maximum number of infections, and improving the long-term epidemic outlook by decreasing the total number of cases. 

In fact, clinically this is quite a common protocol to deal with communicable diseases and their widespread. So, what is the scientific knowledge gap that this manuscript is addressing?

Author Response

We thank the reviewer for their thoughtful comments and constructive suggestions concerning our manuscript entitled “A Compartmental Mathematical Model to Assess the Impact of Vaccination, Isolation, and Key Epidemiological Parameters on Mpox Control” (ID: medsci-3778811), which enabled us to resubmit a clearly improved manuscript. We highlighted the amendments in the revised manuscript, and responded, point by point to, the comments listed below.

Reviewer #1:

Q0. Research article titled, “A Compartmental Mathematical Model to Assess the Impact of 2 Vaccination, Isolation, and Key Epidemiological Parameters on 3 Mpox Control” is an appreciable effort to understand and assess various parameters that could lead or contribute to disease-free state.

As stated by authors, the model suggests that transitioning from a potential endemic state to a disease-free state is achievable through a combination of control measures, such as increasing the vaccination rate (ν), improving vaccine effectiveness (ε), enhancing isolation rate (φ) and reducing immunity loss among vaccinated individuals (κ). These strategies contribute to lowering Râ‚€, reducing the maximum number of infections, and improving the long-term epidemic outlook by decreasing the total number of cases.

R0. We greatly appreciate the comments and suggestions on our manuscript.

Q1. In fact, clinically this is quite a common protocol to deal with communicable diseases and their widespread. So, what is the scientific knowledge gap that this manuscript is addressing?

R1. We thank the reviewer for this valuable observation. While vaccination and isolation are indeed standard public health strategies, the knowledge gap addressed by our study lies in quantifying how these measures interact under realistic conditions of waning immunity and asymptomatic transmission, both of which are underreported but epidemiologically relevant for Mpox. To highlight this in the text, we have added a few lines at the end of the first paragraph of the discussion. The changes are highlighted in yellow.

Reviewer 2 Report

Comments and Suggestions for Authors

Mathematical Model is widely adopted nowadays to study epidemiology of infectious diseases.

While this manuscript is comprehensive, the length is really rather long. The number of tables and figures are excessive and it is suggested to move some to supplementary table and figures.

Also, as monkey pox is mainly transmitted by sexual contact and also a relatively uncommon infection. It means the vulnaerbale population is a restricted group. From public health aspect, to predict the transimission, it may not be advanced Mathematical Model. The auhtors need to justify why using this complciated model is beneficial.

  1. I will also suggest the authors to explain how this model can be implemented to aid public health polices
  2. The authors shall also discuss pros and cons to use mathematical modeling in infectious diseases briefly in discussion session

Author Response

We thank the reviewer for their thoughtful comments and constructive suggestions concerning our manuscript entitled “A Compartmental Mathematical Model to Assess the Impact of Vaccination, Isolation, and Key Epidemiological Parameters on Mpox Control” (ID: medsci-3778811), which enabled us to resubmit a clearly improved manuscript. We highlighted the amendments in the revised manuscript, and responded, point by point to, the comments listed below.

Reviewer #2:

Q0. Mathematical Model is widely adopted nowadays to study epidemiology of infectious diseases.

R0. We greatly appreciate the comments and suggestions on our manuscript.

Q1. While this manuscript is comprehensive, the length is really rather long. The number of tables and figures are excessive and it is suggested to move some to supplementary tables and figures.

R1. We thank the reviewer for this valuable observation. We have reduced the number of figures in the manuscript, transferring several as supplementary material as suggested.

Q2. Also, as monkey pox is mainly transmitted by sexual contact and also a relatively uncommon infection. It means the vulnerable population is a restricted group. From public health aspect, to predict the transmission, it may not be advanced Mathematical Model. The authors need to justify why using this complicated model is beneficial.

R2. We thank the reviewer for this valuable observation. While the 2022 outbreak was concentrated in MSM populations, subsequent evidence showed secondary community transmission and potential zoonotic reintroductions. Therefore, using a more detailed SEIR-based model is justified, as it captures broader transmission scenarios, includes waning immunity and asymptomatic carriers, and provides a flexible framework for both current and future epidemic contexts. We have added a line to the third paragraph of the introduction to consolidate this justification. The changes are highlighted in yellow.

Q3. I will also suggest the authors to explain how this model can be implemented to aid public health polices.

R3. We thank the reviewer for this valuable observation. The model can support public health policies by identifying which epidemiological parameters (e.g., vaccination rate, isolation rate, waning immunity) most influence transmission dynamics. This allows decision-makers to prioritize interventions, optimize vaccination strategies, and anticipate healthcare demand under different scenarios. We have added some lines at the end of the discussions to include these ideas. The changes are highlighted in yellow.

Q4. The authors shall also discuss pros and cons to use mathematical modeling in infectious diseases briefly in discussion session.

R4. We thank the reviewer for this valuable observation. We have added a paragraph at the end of the discussion indicating the advantages and limitations of using mathematical models in infectious diseases, as suggested. The changes are highlighted in yellow.

Reviewer 3 Report

Comments and Suggestions for Authors

The authors present an epidemiological model to discuss Mpox epidemics in human populations when neglecting the infection dynamics in the animal populations from which the Mpox virus of interest originates. Using realistic model parameters, the authors discuss the impacts of various model parameters on the infection dynamics and, in particular, discuss effects of vaccination. Moreover, they fit their model to data from North and South America and Europe and discuss among other things again the beneficial impacts of vaccination. In my opinion, the study is a very comprehensive study and provides both researchers and policy makers with various helpful model-based/data-driven insights.  However, the current version of the manuscript, in my opinion, has a few conceptual and mathematical major mistakes that should be revised. There are also a few minor issues the authors should take care of. 

Major points

1) Conceptual framework and literature review

The conceptual framework presented in the manuscript addresses only a part of the entire relevant epidemiological system for Mpox infections: the human system. The entire relevant epidemiological system consists of the animal system and the human system (in fact, the authors acknowledge in their introduction that Mpox is a zoonotic infectious disease). The authors fail to point out in their introduction that Mpox epidemics in general is modeled with the help of two coupled subsystems: the human and the animal subsystem. For a recent review-like study on SIR and SEIR Mpox human-animal models see Frank (2024). For the Mpox outbreaks in South and North America and Europe that are primarily studied by the authors to focus only on the human subsystem might be an appropriate approach. In contrast, for Mpox outbreaks in Africa this view is a simplification. The following revisions should be made

1a) In the introduction the authors should review the general approach to model Mpox infection outbreaks using human and animal subsystem (e.g. Frank 2024 and references therein). The authors should motivate their approach to focus only on the human system (e.g. as a parsimony approach or because they will present data only for “Western World” Mpox infection waves for which the animal system maybe neglected)

Frank TD (2024) Mathematical analysis of four fundamental epidemiological models for Monkeypox disease outbreaks, Mathematics (MDPI), 12, article 3215

1b) In the discussion section, Section 4, the authors should return to this issue. The authors should point out the limitation of their study to neglect the animal system and speculate about extensions of their work that would take the animal system into account

1c) Line 545 the sentence “The model developed to retrospectively analyze the epidemic dynamics in the most affected continents [3]” should be revised. I understand that the authors motivate this sentence by Ref [3]. However, the sentence is misleading. People in certain regions in Africa live everyday with the risk to catch Mpox from animals. In contrast, Western World people have the luxury to take a plane and fly to an African country and catch Mpox there and bring it back home. Taking this point of view, Africa is the continent that is most affected by Mpox. Having said that the authors just should make sure that it is clear that they give the clear message that all humans are important. Studying Mpox epidemics in Europe and the Americas is important. But studying Mpox epidemics in Africa is important as well.

2) Manuscript structure

The middle part of the manuscript consists of  “section 2: material and methods” and “section 3: results”. However, sections 2.1.1-2.1.4 provide results that should be presented in the results section.

3) Mathematical issues

3.1) Proof of positivity of solutions is incomplete

The proof given in the Supplements is incomplete. The reason for this is that the authors assume that N is positive without showing that N in fact is positive. More precisely, in section 1.1.2 of the Supplementary Methods the authors show that dN/dt=Lambda – mu N – omega(I+Q). If N equals zero and Lambda-omega(I+Q) is smaller than zero, then N becomes negative. If N is negative, the proof that the authors use in section 1.1.1 for S does not work for E. E could become negative. If E is negative, then I and A could also become negative. In other words, if N is positive, then the proof of positivity presented in Section 1.1.1 works well and since all variables are positive, this would imply that N is positive. However, the assumption that N is positive should be motivated in the first place. In the ideal case: a proof should be given. Alternatively, the authors may delete the omega terms in their model and assume that N is constant. Or they delete the proof and state that they consider only conditions for which the solutions are positive.

3.2) Proof of theorem 4 is incorrect because the boundedness theorem gives the wrong kind of inequality for N

In the Supplementary Methods in Sec. 1.1.3.3 the authors try to proof that the endemic fixed point only exists for R0 larger than 1. They derive the inequality (24) which says that R0 must be larger than mu times N divided by Lambda. If the term mu times N divided by Lambda would be larger than (or equal to) 1, then this would imply that R0 must be larger than 1. However, the boundedness theorem (theorem 2) gives an inequality in the opposite direction. N is not larger than something but it is smaller than something. In particular, the term mu time N divided by Lambda is smaller than 1. E.g. let us assume N equals zero, which is consistent with the boundedness theorem. Then R0 could be smaller than 1 and the inequality (24) would still be satisfied.

Note that in the literature frequently the proof is given that if and only if R0 is larger than 1 then the endemic fixed point exists. I think that this is also the case for the model presented by the authors. However, the proof that the authors have presented in 1.1.3.3 is incorrect and should be revised somehow in order to show this.

3.3) Theorem 5 unclear

In the Supplementary Methods in Section 1.1.4.1 the authors nicely illustrate that if R0 is smaller than 1 then the cubic equation (involving the coefficients a0, a1, a2,a3) exhibits only negative eigenvalues. However, I do not see in Section 1.1.4.1 any proof that the cubic equation exhibits at least one positive eigenvalue if R0 is larger than 1. That is, the authors have two options. Option 1: they show more clearly that when R0 is larger than 1 that this implies that the cubic equation exhibits at least one positive eigenvalue (or a pair of complex eigenvalues with positive real part). Option 2: the authors change their Theorem 5. In this case  “locally asymptotically stable if and only if R0 is smaller than 1” should read “locally asymptotically stable if R0 is smaller than 1” That is, delete the “and only if”

4) Data fitting and misleading interpretation of I

The compartment of I describes the symptomatically Mpox infected individuals. The authors use this variable in order to fit data of observed Mpox infected individuals. However, the observed Mpox infected individuals are those individuals who have been diagnosed or identified by health officials. In contrast, the model variable I describes the actual symptomatically infected individuals. In general, this number is larger than the actual observed individuals. In the literature, some studies have fitted the truly infected individuals to the observed infected individuals. However, other studies have introduced another additional compartment: the diagnosed infected individuals. The data given in terms of diagnosed (or observed) infected individuals have then be fitted to that separate compartment. A detailed discussion of this issue can be found in Sec. 3.6 and Chapter 8 of the monograph by Frank (2022). Consequently, the authors should made the following revisions

4.1) Section 2.3.1

State explicitly that the variable y showing up in line is the cumulative function of I(t) in the sense that dy/dt is equal to p times eta time E (if this is the way the authors have computed the cumulative function? if not explain how y is related to I)

Address the issue that the observed cumulative cases are not the true cases such that fitting the model with the help of I(t) to the observed data involves some sort of simplification or approximation

4.2) In the discussion section, section 4, address the limitation that comes with fitting the model to the observed data via the compartment I(t). Address possible solutions (e.g. the models reviewed in Frank 2022 using the aforementioned additional compartment)

Frank, T.D. (2022) COVID-19 Epidemiology and virus dynamics, Springer, Berlin.

Minor points

a) Variable names in abstract should be removed. Reason: the abstract is not the place where variables are defined

b) Line 143, confusing statement: vaccination is assumed to provide total … immunity. In fact, the authors introduce the parameter epsilon that describes the effectivity of vaccination. The manuscript reads like that if epsilon equals 1 then there is total immunity. However, in general, for epsilon smaller than 1 there is not total immunity. In short, the notion of total immunity and the notion of the epsilon parameter seem to be two concepts that are in contradiction with each other.

c) Section 2.3.1: the heading of this section reads “Subsubsection” and should be revised

d) Section 3.1.1

It is unclear how the authors selected the two parameters sets for the analysis presented in Fig. 6. E.g. did the authors use the mid-point values of the ranges presented in Table 1?

e) Line 384: the phrase “Tables may have a footer” probably should be deleted.

f) Supplementary Methods: some equations numbers are incorrect

Section 1.1.4.1 We evaluate the Jacobian matrix at the disease-free equilibrium point given in (7) and substitute the expression for R0 from (8).

Eq. (7) is in fact Eq. (5)

Eq. (8) is in fact Eq. (12)

1.1.4.2 Theorem 6, equation number incorrect

There is no equation (28). Maybe the authors mean Eq. (26)?

Line 197 in the main text should be revised in this context as well

g) Supplementary Methods, Section 1.1.4.1

There are long blank spaces within the equations for a2 and a3 that should be removed

Author Response

We thank the reviewer for their thoughtful comments and constructive suggestions concerning our manuscript entitled “A Compartmental Mathematical Model to Assess the Impact of Vaccination, Isolation, and Key Epidemiological Parameters on Mpox Control” (ID: medsci-3778811), which enabled us to resubmit a clearly improved manuscript. We highlighted the amendments in the revised manuscript, and responded, point by point to, the comments listed below.

Reviewer #3:

Q0. The authors present an epidemiological model to discuss Mpox epidemics in human populations when neglecting the infection dynamics in the animal populations from which the Mpox virus of interest originates. Using realistic model parameters, the authors discuss the impacts of various model parameters on the infection dynamics and, in particular, discuss effects of vaccination. Moreover, they fit their model to data from North and South America and Europe and discuss among other things again the beneficial impacts of vaccination. In my opinion, the study is a very comprehensive study and provides both researchers and policy makers with various helpful model-based/data-driven insights. However, the current version of the manuscript, in my opinion, has a few conceptual and mathematical major mistakes that should be revised. There are also a few minor issues the authors should take care of.

R0. We greatly appreciate the comments and suggestions on our manuscript.

Conceptual framework and literature review. The conceptual framework presented in the manuscript addresses only a part of the entire relevant epidemiological system for Mpox infections: the human system. The entire relevant epidemiological system consists of the animal system and the human system (in fact, the authors acknowledge in their introduction that Mpox is a zoonotic infectious disease). The authors fail to point out in their introduction that Mpox epidemics in general is modeled with the help of two coupled subsystems: the human and the animal subsystem. For a recent review-like study on SIR and SEIR Mpox human-animal models see Frank (2024). For the Mpox outbreaks in South and North America and Europe that are primarily studied by the authors to focus only on the human subsystem might be an appropriate approach. In contrast, for Mpox outbreaks in Africa this view is a simplification. The following revisions should be made

Q1. 1a) In the introduction the authors should review the general approach to model Mpox infection outbreaks using human and animal subsystem (e.g. Frank 2024 and references therein). The authors should motivate their approach to focus only on the human system (e.g. as a parsimony approach or because they will present data only for “Western World” Mpox infection waves for which the animal system maybe neglected).

R1. We thank the reviewer for this valuable observation. We agree with the reviewer that integrating human and animal subsystems is highly valuable to fully capture the zoonotic nature of Mpox, as highlighted by Frank (2024) and others. At the same time, because our study analyzes the 2022–2023 outbreaks in non-endemic regions—driven mainly by human-to-human transmission—we adopted a parsimonious approach restricted to the human system, which best matched the available data as noted. We have added a few lines to the introduction to reflect these ideas and justification. The changes are highlighted in yellow.

Q2. 1b) In the discussion section, Section 4, the authors should return to this issue. The authors should point out the limitation of their study to neglect the animal system and speculate about extensions of their work that would take the animal system into account

R2. We thank the reviewer for this valuable observation. We have now acknowledged in the Discussion that neglecting the animal subsystem is a limitation of our study. We also added a short paragraph outlining how future extensions could incorporate reservoir hosts and cross-species transmission dynamics to better capture the zoonotic nature of Mpox. The changes are highlighted in yellow.

Q3. 1c) Line 545 the sentence “The model developed to retrospectively analyze the epidemic dynamics in the most affected continents [3]” should be revised. I understand that the authors motivate this sentence by Ref [3]. However, the sentence is misleading. People in certain regions in Africa live everyday with the risk to catch Mpox from animals. In contrast, Western World people have the luxury to take a plane and fly to an African country and catch Mpox there and bring it back home. Taking this point of view, Africa is the continent that is most affected by Mpox. Having said that the authors just should make sure that it is clear that they give the clear message that all humans are important. Studying Mpox epidemics in Europe and the Americas is important. But studying Mpox epidemics in Africa is important as well.

R3. We thank the reviewer for this valuable observation. We have revised the sentence to explicitly acknowledge that while our analysis focuses on Europe and the Americas, Mpox remains endemic in Africa, where the zoonotic risk and human impact are substantial and equally important to study. The changes are highlighted in yellow.

Manuscript structure

Q4. The middle part of the manuscript consists of  “section 2: material and methods” and “section 3: results”. However, sections 2.1.1-2.1.4 provide results that should be presented in the results section.

R4. We thank the reviewer for this valuable observation. We have transferred sections 2.1.1-2.1.4 to results as suggested (now 3.1.1-3.1.4). The changes are highlighted in yellow.

Mathematical issues solutions

Q5.1) Proof of positivity of solutions is incomplete

The proof given in the Supplements is incomplete. The reason for this is that the authors assume that N is positive without showing that N in fact is positive. More precisely, in section 1.1.2 of the Supplementary Methods the authors show that dN/dt=Lambda – mu N – omega(I+Q). If N equals zero and Lambda-omega(I+Q) is smaller than zero, then N becomes negative. If N is negative, the proof that the authors use in section 1.1.1 for S does not work for E. E could become negative. If E is negative, then I and A could also become negative. In other words, if N is positive, then the proof of positivity presented in Section 1.1.1 works well and since all variables are positive, this would imply that N is positive. However, the assumption that N is positive should be motivated in the first place. In the ideal case: a proof should be given. Alternatively, the authors may delete the omega terms in their model and assume that N is constant. Or they delete the proof and state that they consider only conditions for which the solutions are positive.

R5. We thank the reviewer for this valuable observation. We have replaced the positivity proof in the Supplementary Methods by a complete argument that (i) treats the  case by continuous extension of the incidence, and (ii) uses the standard boundary-check (first hitting time) argument to show every compartment derivative is nonnegative at the boundary. The new proof is included in the revised Supplementary Methods 1.1.1 and guarantees forward invariance of the nonnegative orthant.

Q6. 3.2) Proof of theorem 4 is incorrect because the boundedness theorem gives the wrong kind of inequality for N

In the Supplementary Methods in Sec. 1.1.3.3 the authors try to proof that the endemic fixed point only exists for R0 larger than 1. They derive the inequality (24) which says that R0 must be larger than mu times N divided by Lambda. If the term mu times N divided by Lambda would be larger than (or equal to) 1, then this would imply that R0 must be larger than 1. However, the boundedness theorem (theorem 2) gives an inequality in the opposite direction. N is not larger than something but it is smaller than something. In particular, the term mu time N divided by Lambda is smaller than 1. E.g. let us assume N equals zero, which is consistent with the boundedness theorem. Then R0 could be smaller than 1 and the inequality (24) would still be satisfied.

Note that in the literature frequently the proof is given that if and only if R0 is larger than 1 then the endemic fixed point exists. I think that this is also the case for the model presented by the authors. However, the proof that the authors have presented in 1.1.3.3 is incorrect and should be revised somehow in order to show this.

R6. We thank the reviewer for this valuable observation. We have replaced the previous argument with a direct algebraic derivation that expresses all compartments as linear functions of , and solves the equilibrium equation explicitly. This completes the proof of existence and uniqueness of the endemic equilibrium if only if .

Q7. 3.3) Theorem 5 unclear

In the Supplementary Methods in Section 1.1.4.1 the authors nicely illustrate that if R0 is smaller than 1 then the cubic equation (involving the coefficients a0, a1, a2,a3) exhibits only negative eigenvalues. However, I do not see in Section 1.1.4.1 any proof that the cubic equation exhibits at least one positive eigenvalue if R0 is larger than 1. That is, the authors have two options. Option 1: they show more clearly that when R0 is larger than 1 that this implies that the cubic equation exhibits at least one positive eigenvalue (or a pair of complex eigenvalues with positive real part). Option 2: the authors change their Theorem 5. In this case  “locally asymptotically stable if and only if R0 is smaller than 1” should read “locally asymptotically stable if R0 is smaller than 1” That is, delete the “and only if”

R7. We thank the reviewer for this valuable observation. We have completed the proof: when  the cubic factor has , so  and  for large , hence a positive real root exists by the intermediate value theorem. We have added this argument to the Supplementary Methods 1.1.4.1, completing the "if and only if" statement.

Q8. 4) Data fitting and misleading interpretation of I

The compartment of I describes the symptomatically Mpox infected individuals. The authors use this variable in order to fit data of observed Mpox infected individuals. However, the observed Mpox infected individuals are those individuals who have been diagnosed or identified by health officials. In contrast, the model variable I describes the actual symptomatically infected individuals. In general, this number is larger than the actual observed individuals. In the literature, some studies have fitted the truly infected individuals to the observed infected individuals. However, other studies have introduced another additional compartment: the diagnosed infected individuals. The data given in terms of diagnosed (or observed) infected individuals have then be fitted to that separate compartment. A detailed discussion of this issue can be found in Sec. 3.6 and Chapter 8 of the monograph by Frank (2022). Consequently, the authors should made the following revisions

R8. We thank the reviewer for pointing out the issue regarding how we implemented the simulation and the model fitting. In the initial version, we directly accumulated the number of infected individuals , which indeed does not correctly represent the procedure for comparing against reported cases (either daily or cumulative), since reported cases correspond to flows of transitions between compartments rather than the stock of individuals in a given compartment. Following the reviewer's recommendation, we have corrected the approach by considering an appropriate flow indicated now in Parameter Estimation and Model Fitting section. In the revised version, we have implemented this correction in the simulation code, which allows fitting and comparing both daily and cumulative case series, consistently grounded on the appropriate flows. The new analysis yielded a more coherent and better-interpreted fit according to the model dynamics. The changes are highlighted in yellow.

Q9. 4.1) Section 2.3.1

State explicitly that the variable y showing up in line is the cumulative function of I(t) in the sense that dy/dt is equal to p times eta time E (if this is the way the authors have computed the cumulative function? if not explain how y is related to I)

Address the issue that the observed cumulative cases are not the true cases such that fitting the model with the help of I(t) to the observed data involves some sort of simplification or approximation

R9. We appreciate the reviewer's insightful observation. In our model, the compartment  describes the actual number of symptomatically infected individuals, while the surveillance data correspond to the number of reported/diagnosed cases. Therefore, the observed cumulative cases do not match exactly the true symptomatic cases. To link our model to the available data, we introduce a reporting factor  and subsequent simplification. This formulation makes explicit that fitting directly with  (assuming  ) is a simplification, while incorporating  provides a more realistic mapping between the model and the observed data. The changes are highlighted in yellow.

Q10. 4.2) In the discussion section, section 4, address the limitation that comes with fitting the model to the observed data via the compartment I(t). Address possible solutions (e.g. the models reviewed in Frank 2022 using the aforementioned additional compartment)

R10. We agree with the reviewer that fitting the compartment  directly to the observed data involves a simplification. The main limitation of this approach is that it ignores underreporting and diagnostic delays, which can bias parameter estimates.

In the revised version, we address this limitation explicitly in the Discussion (Section 4). We note that more advanced models incorporate an additional compartment for "diagnosed" or "reported" infections, which separates the epidemiological reality from the surveillance data (see Frank, 2022, Sec. 3.6). Such extensions may be particularly important when underreporting is severe or time-dependent.

For the present work, we opted for the simpler approach of introducing a constant reporting factor , which allows us to estimate parameters while keeping the underlying model structure and theoretical analysis (basic reproduction number, equilibrium analysis, and stability results) unchanged. Nevertheless, we highlight this as a limitation and point to the aforementioned modeling approaches as promising extensions for future work. The changes are highlighted in yellow.

Q11. a) Variable names in abstract should be removed. Reason: the abstract is not the place where variables are defined

R11. We thank the reviewer for this valuable observation. We have revised the abstract to remove variable symbols and replaced them with descriptive terms, keeping the focus on the results and their epidemiological meaning. The changes are highlighted in yellow.

Q12. b) Line 143, confusing statement: vaccination is assumed to provide total … immunity. In fact, the authors introduce the parameter epsilon that describes the effectivity of vaccination. The manuscript reads like that if epsilon equals 1 then there is total immunity. However, in general, for epsilon smaller than 1 there is not total immunity. In short, the notion of total immunity and the notion of the epsilon parameter seem to be two concepts that are in contradiction with each other.

R12. We thank the reviewer for this valuable observation. We agree that the wording was confusing. We have revised the text to clarify that vaccination does not necessarily confer total immunity, but rather a level of protection proportional to vaccine effectiveness (ε). The changes are highlighted in yellow.

Q13. c) Section 2.3.1: the heading of this section reads “Subsubsection” and should be revised

R13. We thank the reviewer for this valuable observation. We have corrected the subsection title. The changes are highlighted in yellow.

Q14. d) Section 3.1.1

It is unclear how the authors selected the two parameters sets for the analysis presented in Fig. 6. E.g. did the authors use the mid-point values of the ranges presented in Table 1?

R14. We thank the reviewer for this valuable observation regarding the selection of parameter sets for the analysis presented in Fig. 6. Indeed, the values were chosen as the extremes of the intervals reported in Table 1, together with the mid-point of those ranges. In the corrected version of the manuscript, we identified and corrected a minor error in the calculation of the mid-point: in the initial version it was mistakenly given as 0.05004, while the correct value should be 0.05504. Accordingly, Fig. 6C (now Fig. S3A) has been updated to reflect this correction. Importantly, this modification does not alter the qualitative features or overall interpretation of the results.

Q15. e) Line 384: the phrase “Tables may have a footer” probably should be deleted.

R15. We thank the reviewer for this valuable observation. We have added the footer to table 2. The changes are highlighted in yellow.

Q16. f) Supplementary Methods: some equations numbers are incorrect

Section 1.1.4.1 We evaluate the Jacobian matrix at the disease-free equilibrium point given in (7) and substitute the expression for R0 from (8).

Eq. (7) is in fact Eq. (5)

Eq. (8) is in fact Eq. (12)

1.1.4.2 Theorem 6, equation number incorrect

There is no equation (28). Maybe the authors mean Eq. (26)?

Line 197 in the main text should be revised in this context as well

R16. We thank the reviewer for this valuable observation. We have corrected all the incorrect ecuation´s number designations. The changes are highlighted in yellow.

Q17. g) Supplementary Methods, Section 1.1.4.1

There are long blank spaces within the equations for a2 and a3 that should be removed

R17. We thank the reviewer for this valuable observation. We have reduced the spaces within the sections indicated as suggested.

Round 2

Reviewer 3 Report

Comments and Suggestions for Authors

The authors answered all my questions and revised the manuscript in line with my comments and suggestions. In particular, I am happy with the revisions of the theorems (1, 4, 5) in the supplementary materials.